# Gallstone Is Associated with Metabolic Factors and Exercise in Korea

**DOI:** 10.3390/healthcare10081372

**Published:** 2022-07-24

**Authors:** Hoyoung Wang, Hoonsub So, Sung Woo Ko, Seok Won Jung, Sung-Jo Bang, Eun Ji Park

**Affiliations:** 1Department of Internal Medicine, Asan Medical Center, University of Ulsan College of Medicine, Seoul 05505, Korea; docwang90@gmail.com; 2Department of Internal Medicine, Ulsan University Hospital, University of Ulsan College of Medicine, Ulsan 44033, Korea; swjung@uuh.ulsan.kr (S.W.J.); sjbang@uuh.ulsan.kr (S.-J.B.); 3Department of Internal Medicine, Eunpyeong St. Mary’s Hospital, Catholic University of Korea, Seoul 03312, Korea; 4BigData Center, Ulsan University Hospital, University of Ulsan College of Medicine, Ulsan 44033, Korea; 0735779@uuh.ulsan.kr

**Keywords:** gallstones, blood glucose, blood pressure, exercise

## Abstract

Gallstone is a common health problem. Cholesterol stone accounts for 90% of stones in the United States and Europe, but East Asia has a high proportion of pigment stone. The aim of this study was to determine the relationship between modifiable metabolic factors and gallstone in a region with a high prevalence of pigment stone. Among 3159 participants who underwent health screening at Ulsan University Hospital from March 2014 to June 2019, 178 patients were diagnosed with gallstone using abdominal ultrasonography; 2860 participants were selected as a control group. Demographic and laboratory data, and a medical questionnaire were obtained. Hypertension and diabetes mellitus were more prevalent in the gallstone group. Age, waist circumference, systolic blood pressure (SBP) ≥ 140 mmHg, fasting blood glucose, HbA1c ≥ 6.5%, visceral fat index, normal-attenuated muscle area index, and engaging in vigorous exercise for ≥2 days per week were associated with gallstone by univariate analysis. Through multivariate logistic regression analysis, HbA1c ≥ 6.5% (odds ratio (OR) 1.98, 95% confidence interval (CI) 1.31–2.98), and 2 or more days of vigorous exercise per week (OR 0.66, 95% CI 0.45–0.95) remained significant. The association persisted after adjusted analysis for age and sex. HbA1c ≥ 6.5% were positively associated with the gallstone. Vigorous exercise for at least 2 days weekly may be related to a lower risk of gallstone formation.

## 1. Introduction

Gallstone disease is a common health burden in developed countries. In the United States, gallstone develops in 10–20% of adults, resulting in a cost of $6.5 billion annually [1]. The estimated prevalence of gallstone disease ranges from 5.9 to 21.9% of the adult population in the United States and Europe [2]. In Korea, the prevalence of gallstone disease is estimated to be 2–3%, and its incidence is rising alongside an increased frequency of medical checkups, a Westernized diet, and economic growth [3]. Most gallstones are known to be cholesterol stones, accounting for up to 90% in Western countries. However, pigment stones account for more than 50% of the incidence in Eastern countries. Although the proportion of intrahepatic duct and common bile duct stones is decreasing, the proportion of pigment stones in the gallbladder appears to be the same [4,5]. The pathogenesis of gallstones is multifactorial. Cholesterol stones are associated with bile cholesterol supersaturation (e.g., age, female sex, pregnancy, metabolic syndrome, diet and drugs), impaired gallbladder function (e.g., prompt wight loss, starvation and parenteral nutrition), impaired enterohepatic circulation of bile acids from small bowel disease accompanied by malabsorption (e.g., Crohn’s disease and gluten enteropathy), and altered gut microbiota. On the other hand, pigment stones are reported to be associated with hyperbilirubinemia results from hemolysis (e.g., spherocytosis, sickle cell anemia, malaria and hypersplenism), ineffective erythropoiesis, induced enterohepatic cycling of unconjugated bilirubin and anaerobic bacterial infection producing beta-glucuronidase, but their relationship with metabolic factors remains unclear [1,3]. Considering that pigment stones contain 20–30% of a cholesterol component or a greater proportion found in mixed stone, metabolic factors might be associated with pigment stones. Because gallstone characteristics vary from region to region, the metabolic risk factors in the Eastern region could be different from those in Western countries. Physical activity is also known to decrease the risk of symptomatic gallstone by about 30% [6]. We aimed to determine the association of gallstones with metabolic factors and physical activity in the region where pigment stones are prevalent.

## 2. Materials and Methods

### 2.1. Study Population and Measurements

This study was a cross-sectional study to find metabolic factors associated with gallstone in Korea. We collected data from 3159 participants aged 18 years or older who underwent routine health screening at the health promotion center, Ulsan University Hospital, between March 2014 and June 2019. We measured the participants’ height, weight, and waist circumference and calculated their body mass index (BMI) as weight in kilograms divided by the square of height in meters. Blood sampling was performed in the early morning after overnight fasting, and samples were analyzed at the central certified laboratory of Ulsan University Hospital. All subjects were interviewed by filling out a questionnaire about their medical and family history, medication use, alcohol intake, smoking habit, and exercise. Physical activity was evaluated using the validated Korean version of self-administration format from the International Physical Activity Questionnaire, including questions about physical activities for the last 7 days [7]. Vigorous activities were defined as activities that make subjects breathe much harder than normal, such as heavy lifting, digging, aerobics, or fast bicycling. We checked how many days per week the subjects engaged in vigorous physical activities and how much time spent was spent doing these activities each day. More than 20 min of exercise in a day was counted as 1 day. Ultrasonography and coronary computed tomography angiography (CCTA) were also included. We were able to gather information about the participants’ body composition by analyzing the CCTA images (Figure 1). By selecting the transverse image of the L3 vertebra level, we could automatically calculate the area of subcutaneous and visceral fat (−190 to +30 Hounsfield units; HU), low-attenuated muscle (−29 to +29 HU), and normal-attenuated muscle (+30 to +150 HU). We divided this area by BMI and named it as the index of each variable. The existence of gallstones was assessed by ultrasonography. We excluded 121 participants who had undergone cholecystectomy. This study was approved by the Institutional Review Board of Ulsan University Hospital (IRB file No. 2022-02-028).

### 2.2. Statistical Analysis

We compared patients with gallstone with subjects without gallstone. We compared the clinical characteristics of each group using *t* test for continuous variables and chi-squared test for categorical variables. We conducted multivariate logistic regression analysis with significant variables in the univariate analysis to estimate the risk of gallstone with odds ratio (OR) and 95% confidence interval (CI). In addition, an age- and sex-adjusted logistic regression model was used to determine the association between clinical factors and gallstone. All statistical analyses were conducted using R software created y Robert Gentleman and Ross Ihaka in New Zealand, version 4.2.

## 3. Results

### 3.1. Characteristics of Included Patients

Between March 2014 and June 2019, 3159 patients underwent a health checkup at Ulsan University Hospital. We excluded 121 patients as they had undergone cholecystectomy. A total of 178 patients had gallstones, as detected by abdominal ultrasonography. We compared 2860 patients as a control group without gallstone (Figure 2). There was no statistically significant difference in height, weight, or BMI. The gallstone group included more patients with hypertension and diabetes. There was no difference in the diagnosis of dyslipidemia, smoking, or alcohol history (Table 1).

### 3.2. Univariate Analysis of Risk Factors for Gallstone

The univariate analysis showed that age, waist circumference, higher systolic blood pressure (SBP) ≥ 140 mmHg, fasting blood glucose (FBG), HbA1c ≥ 6.5%, visceral fat index, and normal-attenuated muscle area (NAMA) index were significantly higher in the group with gallstone. Participants in the group without gallstone tended to engage in 2 or more days of vigorous exercise per week compared with those diagnosed with gallstone. There was no significant difference in hemoglobin, total bilirubin, direct bilirubin, aspartate aminotransferase, alanine aminotransferase, gamma glutamyltransferase, alkaline phosphatase, lipid profile, subcutaneous fat, or low-attenuated muscle area (LAMA; Table 2).

### 3.3. Multivariate Analysis of Risk Factors for Gallstone

In the multivariate logistic regression analysis, age and HbA1c ≥ 6.5% (OR 1.98; 95% CI: 1.31–2.98) remained significant. Two or more days of weekly vigorous exercise (OR 0.66; 95% CI: 0.45–0.95) decreased the risk of gallstone (Table 3). We applied the age- and sex-adjusted logistic regression model and found that HbA1c ≥ 6.5% (adjusted OR 1.93; 95% CI: 1.28–2.91), and 2 or more days of weekly vigorous exercise (adjusted OR 0.64; 95% CI: 0.44–0.92), remained significant (Table 3).

## 4. Discussion

Because gallstones can sometimes be life-threatening, and there is no effective treatment other than a cholecystectomy, preventing gallstones is important. As age and sex are unmodifiable, we focused in this study on modifiable metabolic factors.

Studies about the risk factors associated with gallstone have been primarily conducted in Western countries, where cholesterol stones account for 90% of the incidence. Gallstone characteristics are different in East Asia, and there have been few studies on the risk factors in this region. According to a study that included male subjects who underwent a retirement health examination in Japan, the waist-to-hip circumference ratio was weakly associated with gallstones [8]. A population-based study conducted in Taiwan found that obesity, hyperlipidemia, hepatitis B and C infection, and cirrhosis increased the risk for gallstone in men and women [9].

In terms of metabolic factors, the gallstone group included patients with hypertension and diabetes. In the univariate analysis, age, waist circumference, SBP ≥ 140 mmHg, FBG, and HbA1c ≥ 6.5% (OR 1.98; 95% CI: 1.31–^2^.98) increased the risk of gallstone. After conducting the multivariate analysis, HbA1c ≥ 6.5% remained a significant factor. The association between gallstone formation and higher HbA1c was found in case-control studies conducted in patients with type ^2^ diabetes mellitus [10,11]. This result implies that metabolic factors, especially HbA1c, are also associated with stones other than cholesterol stones. Therefore, managing these factors may help to prevent stone formation.

We also analyzed subcutaneous fat, visceral fat, LAMA, and NAMA, which have been found to be related to insulin resistance and dyslipidemia, increasing the risk of metabolic syndrome. The NAMA, an area of muscle in which little fatty infiltration occurs, represents an area of good quality muscle. LAMA represents a low-quality muscle area, where high levels of lipid content are found among the muscle fibers. These factors can be analyzed using CCTA images to gather information about body composition. Although the visceral fat index and NAMA index showed an association in the univariate analysis, they did not remain significant after multivariate analysis. In determining the risk of gallstone in real clinical settings, measuring blood pressure, FBS, waist circumference, and HbA1c seems to be more clinically useful.

In terms of lipid profile, surprisingly, no sole factor was associated with gallstone. Previous studies have shown an association between lipid profile and gallstone. In a study of 210 patients diagnosed with primary hyperlipoproteinemia, the incidence of gallstone was within a normal range in patients with type IIa hyperlipoproteinemia and higher in those with type IV hyperlipoproteinemia [12]. In a study including 2,068,523 subjects undergoing a health checkup in China, the authors reported that higher total cholesterol, triglycerides, and low-density lipoprotein cholesterol and lower high-density lipoprotein cholesterol increased the risk of gallstone [13]. The discrepancy could be a matter of ethnicity and study design.

Exercise has been reported to decrease the incidence of symptomatic gallstones. A prior study found that increasing exercise to 30 min of endurance-type training five times per week decreased the risk of symptomatic gallstone by 34% in men [6]. Other research indicated that subjects with a higher physical activity index had a lower risk of gallbladder disease [14,15]. According to our study, engaging in vigorous exercise for more than 2 days per week was associated with a lower risk of gallstone (OR 0.66; 95% CI: 0.45–0.95). The mechanism of the protective effect of exercise has been proposed to be the protection of patients from central and general obesity [16,17] and diabetes [18]; a reduction in total cholesterol, triglycerides [19], and insulin resistance [20]; and an increase in plasma cholecystokinin [21] and vagal tone [22], which stimulates gallbladder contraction and emptying. However, the limitation is a lack of exact quantification of strength and duration of exercise. The definition of vigorous exercise—an activity that makes subjects breathe much harder than normal—is vague. The exact cutoff levels of strength and duration for preventing gallstones should be evaluated in future studies. Even though the preventive role of exercise for gallstone is not clear, vigorous exercise may be encouraged for gallstone prevention.

Our study has several limitations. First, because this study used a cross-sectional design, any causal relationship between stone and risk factors should be interpreted cautiously. Second, we were able to analyze only variables included in the medical checkups, and we could not determine whether gallstone-related symptoms such as recurrent upper abdominal pain or dyspepsia existed. Third, the proportion of stone types in the study was not clear. Fourth, information regarding the participants was gathered by a self-reporting questionnaire, so inaccuracies could exist. Lastly, because the study population included employees who underwent a health checkup done by their companies, there could be a socioeconomical bias.

## 5. Conclusions

In conclusion, the metabolic factor (HbA1c ≥ 6.5%) was positively associated with gallstone, even in an area in which pigment stones are prevalent. Vigorous exercise for at least 2 days per week may be related to a lower risk of gallstone formation.

## Figures and Tables

**Figure 1 healthcare-10-01372-f001:**
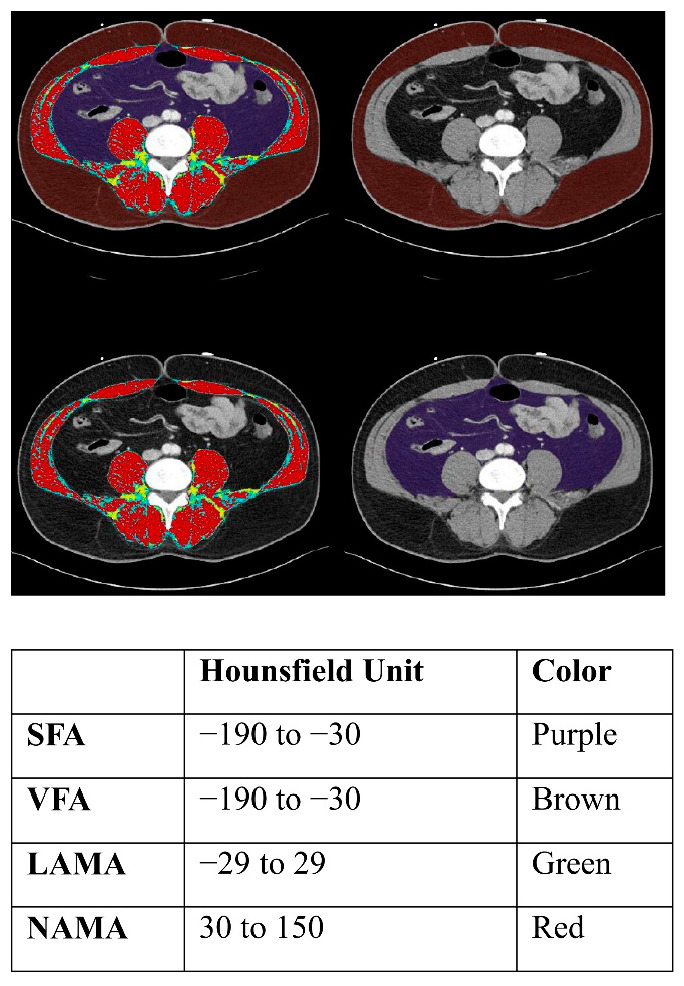
Segmental abdominal body fat and muscle analysis at the L3 vertebra on abdominopelvic computed tomography. SFA, subcutaneous fat area; VRA, visceral fat area; LAMA, low-attenuated abdominal muscle area; NAMA, normal-attenuated muscle area.

**Figure 2 healthcare-10-01372-f002:**
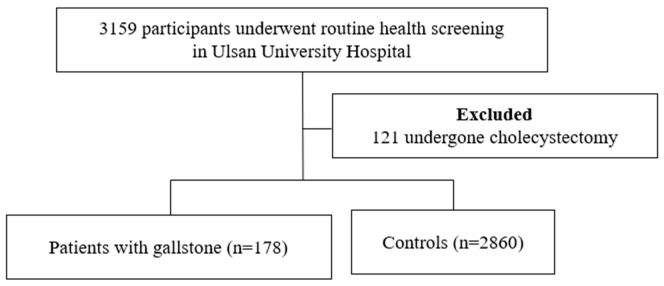
Overview of the study population.

**Table 1 healthcare-10-01372-t001:** Baseline characteristics of study participants.

	No Gallstone (n = 2860)	Gallstone (n = 178)	*p*-Value
Age, mean (SD)	55.1 (8.3)	57.4 (8.5)	<0.001
Gender, no (%)			0.750
Female	1178 (41.2)	76 (42.7)	
Male	1682 (58.8)	102 (57.3)	
Height, cm, mean (SD)	165.1 (8.7)	164.5 (9.3)	0.338
Weight, kg, mean (SD)	66.0 (11.8)	66.6 (11.5)	0.471
BMI, kg/m^2^, mean (SD)	24.1 (3.1)	24.5 (3.1)	0.087
Chronic disease, no (%)			
Hypertension	977 (34.2)	86 (48.3)	<0.001
Diabetes mellitus	366 (12.8)	45 (25.3)	<0.001
Dyslipidemia	447 (15.6)	34 (19.1)	0.260
Smoking, no (%)			0.581
Never smoker	1487 (52.0)	93 (52.2)	
Ex-smoker	758 (26.5)	42 (23.6)	
Current smoker	615 (21.5)	43 (24.2)	
Alcohol consumption, no (%)			0.960
No	1159 (40.5)	73 (41.0)	
Yes	1701 (59.5)	105 (59.0)	

BMI, body mass index.

**Table 2 healthcare-10-01372-t002:** Univariate analysis of risk factors for gallstone.

	No Gallstone (n = 2860)	Gallstone (n = 178)	*p*-Value
Age, mean (SD)	55.1 (8.3)	57.4 (8.5)	<0.001
Gender, no (%)			0.750
Female	1178 (41.2)	76 (42.7)	
Male	1682 (58.8)	102 (57.3)	
Waist circumference, cm, mean (SD)	85.4 (8.1)	87.1 (8.0)	0.007
SBP, no (%)			0.018
≥140 mmHg	335 (11.7)	32 (18.0)	
<140 mmHg	2525 (88.3)	146 (82.0)	
Serum indices			
Hb, g/dL, mean (SD)	14.4 (1.5)	14.3 (1.6)	0.347
Total bilirubin, mg/dL, mean (SD)	0.9 (0.4)	0.8 (0.3)	0.107
Direct bilirubin, mg/dL, mean (SD)	0.3 (0.1)	0.3 (0.1)	0.166
AST, IU/L, mean (SD)	25.8 (13.8)	26.4 (14.0)	0.583
ALT, IU/L, mean (SD)	27.8 (18.2)	30.4 (32.5)	0.277
γGTP, IU/L, mean (SD)	41.3 (52.4)	43.2 (61.6)	0.683
ALP, IU/L, mean (SD)	65.3 (19.6)	66.8 (18.6)	0.333
Total cholesterol, mg/dL, mean (SD)	190.6 (37.8)	187.8 (41.4)	0.334
TG, mg/dL, mean (SD)	112.8 (76.6)	117.4 (64.3)	0.357
HDL-C, mg/dL, mean (SD)	53.7 (15.9)	51.7 (15.6)	0.110
LDL-C, mg/dL, mean (SD)	130.3 (35.0)	127.6 (38.7)	0.335
FBG, mg/dL, mean (SD)	95.4 (22.4)	100.9 (26.0)	0.007
HbA1c, no (%)			<0.001
≥6.5 %	272 (9.5)	34 (19.1)	
<6.5%	2588 (90.5)	144 (80.9)	
CCTA indices			
Sfat index, cm^2^/(kg/m^2^), mean (SD)	5.9 (2.0)	6.2 (2.1)	0.068
Vfat index, cm^2^/(kg/m^2^), mean (SD)	4.5 (2.2)	4.9 (2.3)	0.018
LAMA index, cm^2^/(kg/m^2^), mean (SD)	1.2 (3.8)	1.2 (0.3)	0.881
NAMA index, cm^2^/(kg/m^2^), mean (SD)	4.5 (4.2)	4.2 (1.2)	0.015
Vigorous exercise, no (%)			0.015
Less than 2 days in a week	1996 (69.8)	140 (78.7)	
2 or more days in a week	864 (30.2)	38 (21.3)	
	**Univariate Analysis**		
	**OR (95% CI)**		** *p*-Values**
Age, mean (SD)	1.04 (1.02–1.05)		<0.001
Gender, no (%)			
Female	1		
Male	0.94 (0.69–1.28)		0.692
Waist circumference, cm, mean (SD)	1.02 (1.01–1.04)		0.007
SBP, no (%)			
<140 mmHg	1		
≥140 mmHg	1.65 (1.09–2.43)		0.014
Serum indices			
Hb, g/dL, mean (SD)	0.95 (0.87–1.05)		0.347
Total bilirubin, mg/dL, mean (SD)	0.74 (0.48–1.10)		0.152
Direct bilirubin, mg/dL, mean (SD)	0.40 (0.11–1.37)		0.162
AST, IU/L, mean (SD)	1.00 (0.99–1.01)		0.583
ALT, IU/L, mean (SD)	1.01 (1.00–1.01)		0.075
γGTP, IU/L, mean (SD)	1.00 (1.00–1.00)		0.638
ALP, IU/L, mean (SD)	1.00 (1.00–1.01)		0.332
Total cholesterol, mg/dL, mean (SD)	1.00 (0.99–1.00)		0.334
TG, mg/dL, mean (SD)	1.00 (1.00–1.00)		0.429
HDL-C, mg/dL, mean (SD)	0.99 (0.98–1.00)		0.110
LDL-C, mg/dL, mean (SD)	1.00 (0.99–1.00)		0.335
FBG, mg/dL, mean (SD)	1.01 (1.00–1.01)		0.002
HbA1c, no (%)			
<6.5%	1		
≥6.5%	2.25 (1.49–3.29)		<0.001
CCTA indices			
Sfat index, cm^2^/(kg/m^2^), mean (SD)	1.07 (0.99–1.15)		0.068
Vfat index, cm^2^/(kg/m^2^), mean (SD)	1.08 (1.01–1.15)		0.018
LAMA index, cm^2^/(kg/m^2^), mean (SD)	1.00 (1.00–1.0^2^)		0.968
NAMA index, cm^2^/(kg/m^2^), mean (SD)	0.86 (0.76–0.98)		0.0^2^1
Vigorous exercise, no (%)			
Less than 2 days in a week	1		
2 or more days in a week	0.63 (0.43–0.90)		0.013

SBP, systolic blood pressure; AST, aspartate aminotransferase; ALT, alanine aminotransferase; γGTP, gamma glutamyltransferase; ALP, alkaline phosphatase; FBG, fasting blood glucose; TG, triglyceride; HDL-C, high-density lipoprotein; LDL-C, low-density lipoprotein; CCTA, coronary CT angiography; Sfat, subcutaneous fat; Vfat, visceral fat; LAMA, low-attenuated muscle area; NAMA, normal-attenuated muscle area.

**Table 3 healthcare-10-01372-t003:** Multivariate analysis of risk factors for gallstone.

	Multivariate Analysis (Crude Model)		Multivariate Analysis (Age- and Sex-Adjusted Model)	
	OR (95% CI)	*p*-Values	OR (95% CI)	*p*-Values
Age	1.03 (1.01–1.05)	0.00^2^		
Waist circumference	1.01 (0.97–1.04)	0.636	1.01 (0.98–1.04)	0.693
SBP ≥ 140 mmHg	1.45 (0.96–^2^.18)	0.079	1.39 (0.9^2^–1^2^.09)	0.1^2^1
Serum indices				
HbA1c ≥ 6.5%	1.98 (1.31–^2^.98)	00.001	1.93 (1.^2^8–^2^.91)	
ALT	1.00 (1.00–1.01)	0.396	1.00 (1.00–1.01)	0.^2^99
CCTA indices				
Sfat index	1.0^2^ (0.91–1.13)	0.76^2^	1.06 (0.94–1.18)	0.341
Vfat index	1.04 (0.94–1.14)	0.45^2^	1.03 (0.93–1.13)	0.63^2^
NAMA index	0.89 (0.76–1.04)	0.1^2^7	0.99 (0.90–1.10)	0.9^2^0
Vigorous exercise				
2 or more days in a week	0.66 (0.45–0.95)	0.0^2^6	0.64 (0.44–0.9^2^)	0.018

SBP, systolic blood pressure; ALT, alanine aminotransferase; Sfat, subcutaneous fat; Vfat, visceral fat; NAMA, normal-attenuated muscle area.

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
