# Peer review of "Gallstone Is Associated with Metabolic Factors and Exercise in Korea"

_healthcare, 2022, doi:10.3390/healthcare10081372_

Round 1

Reviewer 1 Report

The review is about the article titled "Gallstone is associated with metabolic factors and exercise in Korea". The study was trying to assess what are the risk factors of gallstone occurence in region with high prevelence of pigmnet stone.

The study was conducted in Korea in 5 years period (between 2014-2019) in one of the Koreans hospital. The study sample was 3159 participants, 18+, with 178 patients diagnosted with gallstone, so the prevelence was around 6% (without excluded participants). Therefore, the study sample was big, however completing the information from only one hospital, so from only one region of this country, have lower potential to form sufficinet conclusions, in my opinion. The article have also some flaws that need to be pointed.

- in lines 46-48 authors are pointing factors which are associated with pigment stones, however they missing few factors, like levels of bilirubine hydrolysis, enzymes like level of beta-glucuronidase (with internal and external sources, like bacterial infections), chronic cholestasis, or other factors, which are occuring in literature

- the literature describing risk of gallstone are from nearly 15 years ago, i would recomment to try to find newer studies.

- why fiture 2 is before figure 1?

- IPAQ have 4 different versions. Long and short one, as well as self administered pr telephone format. The authors did not show which was used by them, and why they used this item.

- The authors qualified vigorous activities as heavy lifting, digging, areobics, fast bicycling. This is a methodological error, mostly beacuse vigorous activities will be different between different people. The key to find if it is vigorous exercise or not is inside such factors like heart rate or oxygen uptake. Obese people will have different values from this in normal weight.

- Qualification of vigorous activities also cant be just narrowed down for few exercises which we think will be intensive or heavy. There are plenty of house activities that can be vigoues, and lack of exercises doesnt mean that people participating in the studies cant do some of vigorous home activities.

- The IPAQ questionaire is a very useful tool, however i dont think that information just from this questionaire can ber valuble for this kind of analysis. No physical test, lack of performance tests, and lack of knowledge of physical performance of this people is huge limitation in this matter.

- Because of 3 points above, the discussion about impact of PA on gallstone occurence as well as conclusions that "exercise for least 2 days per week may be recommended for the prevention of gallstones" is completly premature and improper.

- lines 94-96 need to be corrected - for now it sounds like authors exluded 121 from 178 patients (which would give us a 57 patients with gallstone), and the figure 1 is showing otherwise.

- The sentence in lines 133-134 is not accurate. The women have no higher risk of gallstone because they are women. Higher prevelence of gallstones are mostly because hormones. Women who gave birth multiply have higher risk of gallstone occurance, the women who started early hormonal contraception, the women who had hormone replacement therapy have higher risk as well. And those are midifialble factors.

- also, did authors ask in their questionaire about medications taken by participants? or about birth control method?

- lines 148-149 - authors cant really say that, beacuse they did not check the gallstone type in patients. its only assumption, that in country with higher prevelence of pigment stones their group have also high prevelence of this stones. Especialy, that most of the results obtained by them are pointing in cholesterol stones direction

Because of the abvove, and many limitation that authors are pointing, i don`t think, that the article is good enought to be published in such good journal with high IF like Healthcare. The study have some potential and useful information to be published elsewhere, however it will require some hard work from atuhors. At the end i would like to point out, that there is no information about ethical comitee agreement or agreement from patients to participate in this study, which is a ethical concern.

Author Response

Response to Reviewer #1

Thank you for your review of our paper. We have answered each of your points below.

Comment 1: in lines 46-48 authors are pointing factors which are associated with pigment stones, however they missing few factors, like levels of bilirubine hydrolysis, enzymes like level of beta-glucuronidase (with internal and external sources, like bacterial infections), chronic cholestasis, or other factors, which are occuring in literature

Response: Authors revised the sentence with pathophysiologic factors contribute to pigment stones formation with clinical examples. Additional citations are marked below. (page #1–2, lines #51–#56)

[On the other hand, pigment stones are reported to be associated with hyperbilirubinemia results from hemolysis (e.g. spherocytosis, sickle cell anemia, malaria and hypersplenism), ineffective erythropoiesis and induced enterohepatic cycling of unconjugated bilirubin, anaerobic bacterial infection producing beta-glucuronidase, but their relationship with metabolic factors remains unclear.]

Vítek L, Carey MC. New pathophysiological concepts underlying pathogenesis of pigment gallstones. Clin Res Hepatol Gastroenterol. 2012;36(2):122-129.

Comment 2: the literature describing risk of gallstone are from nearly 15 years ago, i would recomment to try to find newer studies.

Response: Authors modified the sentence explaining the risk factors for cholesterol stone as pathophysiological factors with relevant clinical examples as the reviewer’s comment #1 with newer studies citated below.  (page #1–2, lines #45–#50)

[Cholesterol stones are associated with bile cholesterol supersaturation (e.g. age, female sex, pregnancy, metabolic syndrome, diet and drugs), impaired gallbladder function (e.g. prompt wight loss, starvation and parenteral nutrition), impaired enterohepatic circula-tion of bile acids from small bowel disease accompanied by malabsorption (e.g. Crohn’s disease and gluten enteropathy), and altered gut microbiota.]

E S, Srikanth MS, Shreyas A, et al. Recent advances, novel targets and treatments for cholelithiasis; a narrative review. Eur J Pharmacol. 2021;908:174376.

Reshetnyak VI. Concept of the pathogenesis and treatment of cholelithiasis. World J Hepatol 2012; 4(2): 18-34.

Comment 3: why figure 2 is before figure 1?

Response: Thank you. We reordered figures in order and corrected the main text according to following orders. (page #2, lines #83, page #3, line #92, and page #3, line #108 and page #4, line #113)

[We were able to gather information about the participants’ body composition by analyz-ing the CCTA images (Figure 1).]

[Figure 1. Segmental abdominal body fat and muscle analysis at the L3 vertebra on abdom-inopelvic computed tomography.]

[We compared 2860 patients as a control group without gallstone (Figure 2).]

[Figure 2. Overview of the study population.]

Comment 4: IPAQ have 4 different versions. Long and short one, as well as self administered pr telephone format. The authors did not show which was used by them, and why they used this item.

Response: Authors used short last 7 days self-administered format of the International Physical Activity Questionnaires (IPAQ), because this method is widely used, cost-effective to assess the usual pattern of physical activities of a large target population, also easily and quickly accessible by study population. We added sentence to reflect the reviewer’s comment in the method section. (page #2, lines #75–#76 )

[Physical activity was evaluated using the validated Korean version of short last 7 days self-administered format of the International Physical Activity Questionnaire.]

Comment 5: The authors qualified vigorous activities as heavy lifting, digging, areobics, fast bicycling. This is a methodological error, mostly beacuse vigorous activities will be different between different people. The key to find if it is vigorous exercise or not is inside such factors like heart rate or oxygen uptake. Obese people will have different values from this in normal weight.

Qualification of vigorous activities also cant be just narrowed down for few exercises which we think will be intensive or heavy. There are plenty of house activities that can be vigoues, and lack of exercises doesnt mean that people participating in the studies cant do some of vigorous home activities.

The IPAQ questionaire is a very useful tool, however i dont think that information just from this questionaire can ber valuble for this kind of analysis. No physical test, lack of performance tests, and lack of knowledge of physical performance of this people is huge limitation in this matter.

Because of 3 points above, the discussion about impact of PA on gallstone occurence as well as conclusions that "exercise for least 2 days per week may be recommended for the prevention of gallstones" is completly premature and improper.

Response: Thank you for the comment. The referee is right to point out that there was lack of individualized and exact qualification of vigorous activities in this study. Also, authors agree with the reviewer that it is premature and improper to state that at least 2 days per week of vigorous exercise may prevent gallstones. Nevertheless, authors would like to appeal that this study is done based on a retrospective design using data from routine health screening. We revised the sentence about the relationship between exercise and gallstone formation in the section of discussion and conclusion as below. (page #7, lines #193–#200 and page #8, lines #213–#215)

[ However, the limitation is lack of exact quantification of strength and duration of exercise. The definition of vigorous exercise -an activity that makes subjects breathe much harder than normal- is vague. The exact cutoff levels of strength and duration for preventing gallstones should be evaluated in the future studies. Even though the preventive role of exercise for gallstone is not clear, vigorous exercise may be encouraged for gallstone prevention.]

[In conclusion, metabolic factor (HbA1c 6.5%) was positively associated with gallstone even in an area in which pigment stones are prevalent. Vigorous exercise for least 2 days per week may be related to lower risk of gallstone formation..]

Comment 6: lines 94-96 need to be corrected - for now it sounds like authors exluded 121 from 178 patients (which would give us a 57 patients with gallstone), and the figure 1 is showing otherwise.

Response: Corrected. We repositioned the sentence forward to make the flow of study more clarifying as the reviewer’s comment. (page #3, lines #104–#106)

[Between March 2014 and June 2019, 3159 patients underwent a health checkup at Ulsan University Hospital. We excluded 121 patients as they had undergone cholecystec-tomy. A total of 178 patients had gallstones as detected by abdominal ultrasonography.]

Comment 7: The sentence in lines 133-134 is not accurate. The women have no higher risk of gallstone because they are women. Higher prevelence of gallstones are mostly because hormones. Women who gave birth multiply have higher risk of gallstone occurance, the women who started early hormonal contraception, the women who had hormone replacement therapy have higher risk as well. And those are midifialble factors.

Response: We agree with the reviewer that there are modifiable women factors for gallstone formation. However, we could not gather information about the women factors such as a history of delivery or medical history about hormonal contraception or hormone replacement therapy on a study setting of routine health check-up. Therefore, authors considered sex as an unmodifiable factor.

Comment 8: also, did authors ask in their questionaire about medications taken by participants? or about birth control method?

Response: The authors asked participants about whether they take medication for hypertension, diabetes, dyslipidemia, or infarction, however, survey was not done about birth control method.

Comment 9: lines 148-149 - authors cant really say that, beacuse they did not check the gallstone type in patients. its only assumption, that in country with higher prevelence of pigment stones their group have also high prevelence of this stones. Especially, that most of the results obtained by them are pointing in cholesterol stones direction.

Response: It is true that authors could not check the type of gallstones, because we could not analyze the exact composition of gallstones only detected in ultrasonography. But, according to the lasted large-volume Korean center reviews as we cited reference #3 and #4, pigment stones remained common in Korea, no distinct increase in cholesterol stones.

Please examine our attachment. Thank you again for reviewing our manuscript in detail and providing helpful comments. We hope that our responses and the corresponding revisions are satisfactory.

Reviewer 2 Report

This study examined metabolic-related risk factors for gallstone using data collected from health checkup in a Korean hospital. The authors claimed, based on their statistical analyses, that the occurrence of gallstone was associated with poor exercise habits, hypertension (high SBP), and high HbA1c.

Main critique:

1)     The interpretation of odds ratio (OR) for SBP and HbA1c is problematic in this paper (Tables 2&3). From univariate or multivariate analyses in these two tables, an odds ratio of 1.01 or 1.02 with p<0.05 (such as that for high SBP) should not be deemed as a significant factor contributing to gallstone. The authors should also understand the meaning of 95% CI for OR. In the second part of Table 2, the OR for high SBP is 1.02 but with 95%CI between 1.01-1.03, considering that the lower CI is smaller than the OR, this is not a significant OR for high SBP. From my assessment of your Tables 2-3, only the exercise habit is significantly inversely associated with the occurrence of gallstone. The association between HbA1c and gallstone is weak but with some significance. From your Tables, all other metabolic factors (including high SBP) are not significantly associated with gallstone. Therefore, the conclusion and many other parts of this manuscript should be revised.

2)     The authors should specify the definition of hypertension and the cutoffs of SBP/DBP in the method section. If only high SBP was used in the logistic analyses, the authors should not simply put “SBP” in the Tables but specify this term.

3)     Was this study approved by IRB or the hospital ethic committee? The statement should be included in the Method section.

4)     There are parts of the writing not clearly stated. Lines 66-67: “breathe much harder than normal”; is there an objective criterion for it? “digging” in Line 67 is also not a clear term.

5)     LINE 49: “cholesterol consists of 20-30% pigment stone..”;  I think you mean the other way around.

6)     LINE 184: the first limitation is not necessary.

7)     LINE 188: gallstone-related symptoms, such as..?  This should be included in the Introduction section.

8)     LINE 192: It is unclear why health checkup is affected by socioeconomic status. The authors should briefly explain this in Korea.

Author Response

Response to Reviewer #2

Thank you for reviewing our manuscript. Our answers to your queries are as follows.

Comment 1: The interpretation of odds ratio (OR) for SBP and HbA1c is problematic in this paper (Tables 2&3). From univariate or multivariate analyses in these two tables, an odds ratio of 1.01 or 1.02 with p<0.05 (such as that for high SBP) should not be deemed as a significant factor contributing to gallstone. The authors should also understand the meaning of 95% CI for OR. In the second part of Table 2, the OR for high SBP is 1.02 but with 95%CI between 1.01-1.03, considering that the lower CI is smaller than the OR, this is not a significant OR for high SBP. From my assessment of your Tables 2-3, only the exercise habit is significantly inversely associated with the occurrence of gallstone. The association between HbA1c and gallstone is weak but with some significance. From your Tables, all other metabolic factors (including high SBP) are not significantly associated with gallstone. Therefore, the conclusion and many other parts of this manuscript should be revised.

Comment 2: The authors should specify the definition of hypertension and the cutoffs of SBP/DBP in the method section. If only high SBP was used in the logistic analyses, the authors should not simply put “SBP” in the Tables but specify this term.

Response:  Thank you for your comment. Authors agree with the reviewer that odds ratio of SBP and HbA1c should show more strong significance, and there should be specific definition with cutoffs of each variable. Therefore, we categorized the participants into two groups by cutoff value 140 mmHg and 6.5% for SBP and HbA1c, respectively. The odds ratio was 1.65 with p<0.014 for SBP ≥ 140 mmHg and 2.25 with p<0.001 for HbA1c ≥ 6.5% in univariate analysis. HbA1c remained significant in multivariate analysis, by odds ratio 1.93 with p=0.018, however, SBP ≥ 140 mmHg did not. Although SBP ≥ 140 mmHg did not remain significant in multivariate analysis, the result showed stronger relationship with gallstone. Authors revised the table and paragraphs with the result, discussion and conclusion section. Citations are documented below.

Unger, T., Borghi, C., Charchar, F., Khan, N. A., Poulter, N. R., Prabhakaran, D., ... & Schutte, A. E. (2020). 2020 International Society of Hypertension global hypertension practice guidelines. Hypertension, 75(6), 1334-1357.

Hur, K. Y., Moon, M. K., Park, J. S., Kim, S. K., Lee, S. H., Yun, J. S., ... & Ko, S. H. (2021). 2021 Clinical Practice Guidelines for Diabetes Mellitus in Korea. Diabetes & Metabolism Journal, 45(4), 461-481.

Comment 3: Was this study approved by IRB or the hospital ethic committee? The statement should be included in the Method section.

Response: Yes, this study was approved by IRB. As the reviewer’s comment, we added a sentence to clarify the approval in the the method section. (page #2, lines #89–#90)

[This study was approved by the Institutional Review Board of Ulsan University Hospital (IRB file No. 2022-02-028).]

Comment 4: There are parts of the writing not clearly stated. Lines 66-67: “breathe much harder than normal”; is there an objective criterion for it? “digging” in Line 67 is also not a clear term.

Response: Authors used short last 7 days self-administered format of the International Physical Activity Questionnaires (IPAQ), because of its cost-effectiveness and accessibility as commented to Reviewer 1, comment 4. Although authors utilized an international format, there isn’t any objective criterion for “breathe much harder than normal” and “digging” is used as an example for vigorous activity. We added sentence to clarify the version of IPAQ in the method section. (page #2, lines #75–#76 )

[Physical activity was evaluated using the validated Korean version of short last 7 days self-administered format of the International Physical Activity Questionnaire.]

Comment 5: LINE 49: “cholesterol consists of 20-30% pigment stone..”;  I think you mean the other way around.

Response: Gallstones can be classified into either cholesterol stones or pigment stones by their composition. Authors got the concept about the possible relationship between pigment stones and metabolic factors from the fact that cholesterol component also exist in pigment stones. We revised the sentence to reflect our intention refer to the reviewer’s comment. (page #2, lines #57–#59)

[Considering that pigment stones contain 20-30 % of cholesterol component or greater proportion found in mixed stone, metabolic factors might be associated with pigment stones.]

Comment 6: LINE 184: the first limitation is not necessary.

Response: We deleted the first limitation according to reviewer’s comment. (page #8, lines #202–#203)

Comment 7: LINE 188: gallstone-related symptoms, such as..?  This should be included in the Introduction section.

Response: As the reviewer commented, we added examples of gallstone-related symptoms in the sentence. (page #8, lines #206–#207)

[Third, we were able to analyze only variables included in the medical checkups, and we could not determine whether gallstone-related symptoms such as recurrent upper ab-dominal pain or dyspepsia existed.]

Comment 8: LINE 192: It is unclear why health checkup is affected by socioeconomic status. The authors should briefly explain this in Korea.

Response: The study population included employees who underwent health checkup conducted by their companies, therefore the authors thought there could be bias comes from socioeconomic status. We corrected the sentence by adding brief explanation. (page #8, lines #210–#211)

[Lastly, because the study population included employees who underwent health checkup done by their companies, there could be a socioeconomical bias.]

Please examine our attachment. Thank you again for reviewing our manuscript in detail and providing helpful comments. We hope that our responses and the corresponding revisions are satisfactory.

Round 2

Reviewer 2 Report

The revision is much improved. I only have a comment about the strange name of the questionnaire (translated from Korean)- short 7 days?

Please have it translated properly 

Author Response

Response to Reviewer #2

Thank you for reviewing our manuscript. Our answers to your queries are as follows.

Comment: I only have a comment about the strange name of the questionnaire (translated from Korean)- short 7 days? Please have it translated properly

Response:  Thank you for your comment. Although authors used the official name of the questionnaire, we agree that it will be awkward while reading the manuscript. Authors changed the sentence to be more fluent to read. (page #2, lines #75–#77 )

[Physical activity was evaluated using the validated Korean version of self-administration format from the International Physical Activity Questionnaire, including questions about physical activities for the last 7 days.]

Thank you again for reviewing our manuscript in detail and providing helpful comments. We hope that our responses and the corresponding revisions are satisfactory. Please see the attachment.
